# DWV 3C Protease Uncovers the Diverse Catalytic Triad in Insect RNA Viruses

Xuye Yuan,[a] Tatsuhiko Kadowaki[a]

[a]Department of Biological Sciences, Xi'an Jiaotong-Liverpool University, Jiangsu Province, China

**ABSTRACT** Deformed wing virus (DWV) is the most prevalent Iflavirus that is infecting honey bees worldwide. However, the mechanisms of its infection and replication in host cells are poorly understood. In this study, we analyzed the structure and function of DWV 3C protease ($3C^{pro}$), which is necessary for the cleavage of the polyprotein to synthesize mature viral proteins. Thus, it is one of the nonstructural viral proteins essential for the replication. We found that the $3C^{pro}$s of DWV and picornaviruses share common enzymatic properties, including sensitivity to the same inhibitors, such as rupintrivir. The predicted structure of DWV $3C^{pro}$ by AlphaFold2, the predicted rupintrivir binding domain, and the protease activities of mutant proteins revealed that it has a Cys-His-Asn catalytic triad. Moreover, $3C^{pro}$s of other Iflaviruses and Dicistrovirus appear to contain Asn, Ser, Asp, or Glu as the third residue of the catalytic triad, suggesting diversity in insect RNA viruses. Both precursor $3C^{pro}$ with RNA-dependent RNA polymerase and mature $3C^{pro}$ are present in DWV-infected cells, suggesting that they may have different enzymatic properties and functions. DWV $3C^{pro}$ is the first $3C^{pro}$ characterized among insect RNA viruses, and our study uncovered both the common and unique characteristics among $3C^{pro}$s of *Picornavirales*. Furthermore, it would be possible to use the specific inhibitors of DWV $3C^{pro}$ to control DWV infection in honey bees in future.

**IMPORTANCE** The number of managed honey bee (*Apis mellifera*) colonies has considerably declined in many developed countries in the recent years. Deformed wing virus (DWV) vectored by the mites is the major threat to honey bee colonies and health. To give insight into the mechanism of DWV replication in the host cells, we studied the structure–function relationship of 3C protease ($3C^{pro}$), which is necessary to cleave a viral polyprotein at the specific sites to produce the mature proteins. We found that the overall structure, some inhibitors, and processing of $3C^{pro}$ are shared between *Picornavirales*; however, there is diversity in the catalytic triad. DWV $3C^{pro}$ is the first viral protease characterized among insect RNA viruses and reveals the evolutionary history of $3C^{pro}$ among *Picornavirales*. Furthermore, DWV $3C^{pro}$ inhibitors identified in our study could also be applied to control DWV in honey bees in future.

**KEYWORDS** deformed wing virus, 3C protease, catalytic triad

**Ad Hoc Peer Reviewer** Eugene V. Ryabov

Address correspondence to Tatsuhiko Kadowaki, Tatsuhiko.Kadowaki@xjtlu.edu.cn.

The authors declare no conflict of interest.

Winter colony loss in honey bees is strongly correlated with the presence of deformed wing virus (DWV) and the ectoparasitic mites, *Varroa destructor* and *Tropilaelaps mercedesae* (1–3). The mites transmit DWV to honey bee (3–6) and increase viral loads (6–10). With the spread of *V. destructor*, DWV has become the most prevalent virus that is infecting honey bees worldwide (11). Honey bees often show multiple symptoms due to high DWV levels, including the death of pupae, deformed wings, shortened abdomen, and reduced life span (4, 5, 12, 13).

DWV belongs to the genus Iflavirus in the order *Picornavirales*. It has a nonenveloped virion approximately 30 nm in diameter and contains a positive-strand RNA

genome. The RNA genome is translated into a polyprotein that is processed by the 3C protease (3C$^{pro}$) to produce mature viral proteins (14). The crucial 3C$^{pro}$s have been well characterized in picornaviruses infecting vertebrates and share similar structures and functions (15–21). Indeed, all of them have a Cys-His-Asp–Glu (Asp or Glu) catalytic triad. 3C$^{pro}$ plays important roles in the viral life cycle and host-virus interactions. For example, 3C$^{pro}$ not only processes a viral polyprotein but also cleaves the specific host proteins necessary for transcription, translation, and nucleocytoplasmic trafficking to modify the cell physiology for viral replication. Furthermore, 3C$^{pro}$ and its precursor with RNA-dependent RNA polymerase (RdRP) bind to the 5'UTR region of the viral genome RNA to initiate replication (22, 23). Based on structural conservation studies, inhibitors of picornavirus 3C$^{pro}$s have been identified and characterized to develop potential antiviral treatments (24).

DWV is the best characterized virus among honey bee viruses; however, very little is known about its mechanism of infection and replication in host cells. Because 3C$^{pro}$ of insect RNA virus has never been studied, we expressed and purified DWV 3C$^{pro}$ as a glutathione *S*-transferase (GST) fusion protein to characterize the protease activity and the inhibitors in this study. We also studied the structure–function relationship of DWV 3C$^{pro}$ based on the structure predicted by AlphaFold2 as well as the protease activities of mutant proteins and compared to that of the 3C$^{pro}$s of other Iflaviruses and Dicistrovirus. We focused on characterizing the catalytic triad since it represents the most critical domain of picornavirus 3C$^{pro}$. We further discussed the mechanism of synthesis and maturation of DWV 3C$^{pro}$ in infected cells. Our study on DWV 3C$^{pro}$ gives insight into the mechanism of viral replication and may help to control DWV infection in honey bees using the specific inhibitor.

## RESULTS

**Enzymatic properties of DWV 3C$^{pro}$.** We expressed DWV 3C$^{pro}$ as a GST-fusion protein and further included the potential VPg region (2094-2353 of DWV polyprotein) to make it soluble in *E. coli*. Protease activity of DWV 3C$^{pro}$ was measured by a fluorescence resonance energy transfer (FRET) assay using a 15 amino acid peptide (PVQ<u>AKPEMD</u>NPNPGE) with the cleavage site (25) as the substrate. Six amino acid sequence, AKPEMD, is present at three sites in the DWV polyprotein. To determine the optimum condition to measure the protease activity, we found that the optimum pH for protease activity is 6 (Fig. 1A), and the optimum temperature is between 25 and 35℃ (Fig. 1B). Dithiothreitol was required for maximum activity as a cysteine protease (Fig. 1C) (26). The protease activity (RFU/min) was proportional to the amount of GST-DWV 3C$^{pro}$ added in the reaction mixture (Fig. 1D). Based on the Michaelis-Menten equation, Km, Vmax, and Kcat/Km values were $4.233 \pm 0.256$ $\mu$M, $0.2538 \pm 0.0067$ $\mu$M min$^{-1}$, and $59.96 \pm 2.032$ mM$^{-1}$ min$^{-1}$, respectively (Fig. 1E).

**Inhibitors of DWV 3C$^{pro}$.** We next tested the effects of known protease inhibitors of picornaviruses and other viruses on DWV 3C$^{pro}$. Eight compounds inhibited enzyme activity with various IC$_{50}$ values, as shown in Table 1. Fig. 2 shows the dose-response curves for rupintrivir (27), ebselen (28), disulfiram (29), GC376 (30), carmofur (29, 31), 6,7-dichloroquinoline-5,8-dione (32), quercetin (33), and zinc. Rupintrivir was the most effective compound with the IC$_{50}$ value of 0.36 $\mu$M. Ebselen, disulfiram, and carmofur were only effective without dithiothreitol, as they modified cysteine in the catalytic triad (29). DWV 3C$^{pro}$ is sensitive to some protease inhibitors of mammalian viruses, suggesting that their catalytic domains and substrate-binding domains share similar structures.

We previously showed that rupintrivir suppresses DWV replication in honey bee pupal cells based on the synthesis of RdRP (34). Thus, we tested the effect of ebselen on DWV replication. Ebselen decreased the synthesis of the 97 kDa precursor of RdRP and 3C$^{pro}$ in a concentration-dependent manner (Fig. 3A and B). However, cell viability was reduced at concentrations of 100 and 250 $\mu$M (Fig. 3C), suggesting that ebselen affects the functions of host proteins as well.

**Predicted structure of DWV 3C$^{pro}$ by AlphaFold2.** We used AlphaFold2 to predict the structure of DWV 3C$^{pro}$ (polyprotein 2199–2353) (35, 36). We obtained multiple

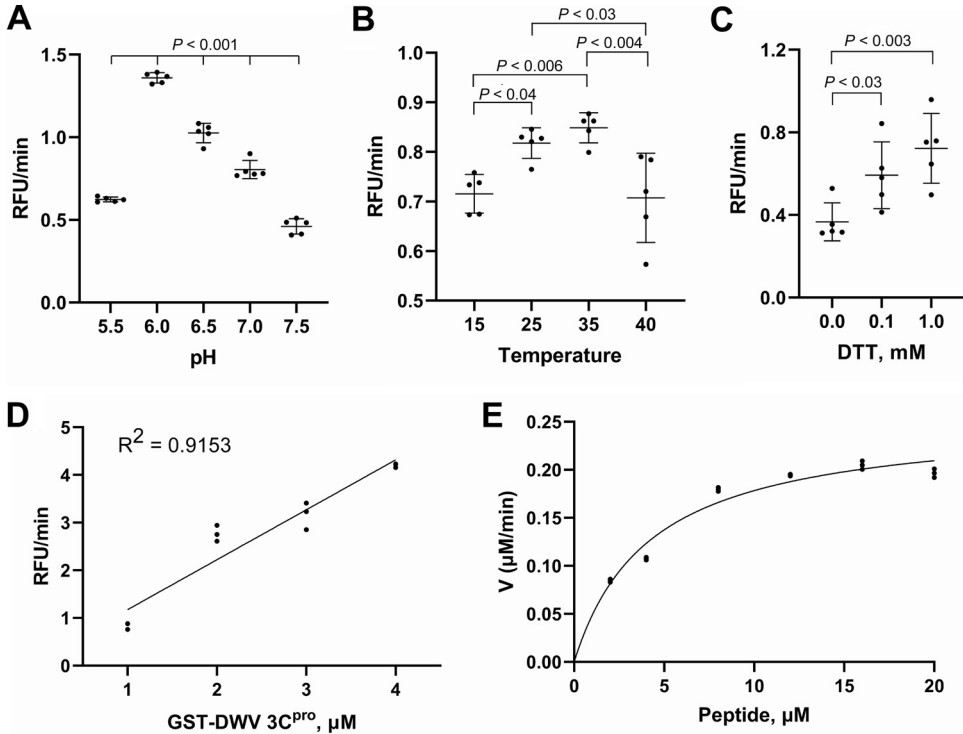

**FIG 1** Enzymatic properties of the deformed wing virus (DWV) 3C protease (3C^pro). (A) Optimum pH for protease activity. Multiple comparisons were made by the Tukey-Kramer method and the *P*-value between pH 6 and other tested pH was < 0.001. (B) Optimum temperature for protease activity. Multiple comparisons were made by the Tukey-Kramer method. Pairs with statistically significant differences are indicated by the *P*-values. (C) Dithiothreitol (DTT) requirement for protease activity. Multiple comparisons were made by the Dunnett method (one-tailed). Mean values ± standard deviation (SD) (error bars) are shown (*n* = 5) for A-C. (D) Protease activity with the indicated concentrations of glutathione *S*-transferase (GST)-DWV 3C^pro protein. (E) Protease activity (V, $\mu$M/min) at the indicated peptide substrate concentrations (2-20 $\mu$M). Mean values ± SD (error bars) are shown (*n* = 3) for D and E.

models and one of them with the overall structure is shown in Fig. 4A. Based on the IDDT values per position, the structure was predicted with high confidence, except for the N- and C termini as well as single loop (polyprotein 2330–2332) (Fig. S1 in the supplemental material). DWV 3C^pro has a trypsin-like structure consisting of six $\beta$-sheets, which are folded into two $\beta$-barrel domains packed perpendicularly to each other. The catalytic center appears to be present between the two domains. This is similar to those of other 3C^pros in addition to the position of $\beta$-ribbon (polyprotein 2281–2293) which constitutes the substrate-binding domain (15–21) (Fig. 4A). To identify the amino acids critical for protease activity, we aligned 3C^pro sequences of DWV and 23 other Iflaviruses (Fig. S2 in the supplemental material). Among the conserved 12 amino acids, the positions of C2307 and H2170 together with nonconserved N2227 in the DWV 3C^pro structure corresponded to those of the known 3C^pro catalytic triads (15–21). Although Y2171 was well conserved, its position relative to H2170 did not appear to be

**TABLE 1** IC$_{50}$ values of DWV 3C^pro inhibitors

| Compound | IC$_{50}$ ($\mu$M) |
|---|---|
| Rupintrivir | 0.36 ± 0.03 |
| Ebselen | 1.66 ± 0.02 |
| Disulfiram | 35.6 ± 0.8 |
| GC376 | 55.2 ± 4.0 |
| Carmofur | 126.7 ± 6.2 |
| 6,7-dichloroquinoline-5,8-dione | 129.3 ± 1.8 |
| Quercetin | 430 ± 62.8 |
| Zinc | 15240 ± 422 |

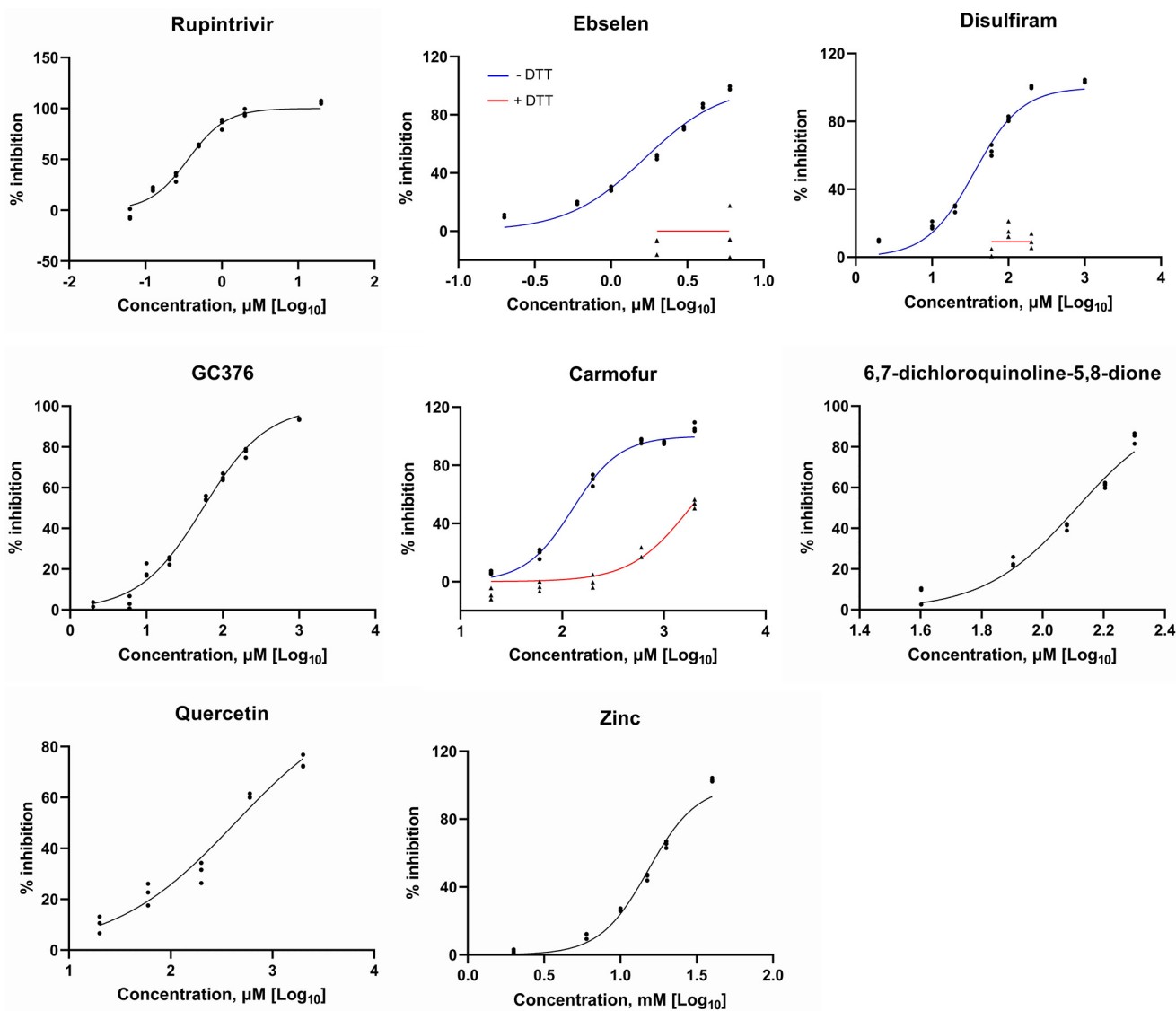

**FIG 2** Compounds that inhibit the activity of the deformed wing virus (DWV) 3C protease (3C^pro). Protease activity of DWV 3C^pro in the presence of increasing concentrations of rupintrivir, ebselen, disulfiram, GC376, carmofur, 6,7-dichloroquinoline-5,8-dione, quercetin, and zinc. Ebselen, disulfiram, carmofur were tested in the absence (blue line) or presence (red line) of 2 mM dithiothreitol (DTT). Other compounds were tested in the presence of DTT. All data ($n = 3$) are shown at the indicated concentrations.

compatible with the third residue of the catalytic triad. There is also D2225, but the side chain with a carboxyl group was oriented in the opposite direction (Fig. 4C and D). The predicted distances between C2307 and H2170 as well as H2170 and N2227 were 4.03 Å and 3.22 Å, respectively in range of interaction (Fig. 4B). We also predicted the 3C^pro structures of four different Iflaviruses: *Brevicoryne brassicae* virus (BrBV), *Laodelphax striatellus* picorna-like virus 2 (LsPVl2), Sacbrood virus (SBV), and *Spodoptera exigua* iflavirus (SeIV-1), and one Dicistrovirus, Cricket paralysis virus (CrPV). For the catalytic triads, all of them had Cys and His residues; however, BrBV, LsPVl2, and SeIV-1 appeared to contain Asn, Ser, and Glu, respectively, as the third residue. SBV and CrPV appeared to have Asp (Fig. S3 in the supplemental material). Thus, 3C^pro of insect RNA virus is likely to have invariant Cys as well as His and various amino acid with side chain containing oxygen as the catalytic triad.

To further examine the structure of DWV 3C^pro predicted by AlphaFold2, we prepared mutant proteins in which the potentially critical amino acid was substituted with alanine and measured the protease activities. The activities were dramatically reduced with the mutant proteins for the catalytic triad (C2307, H2170, and N2227) and

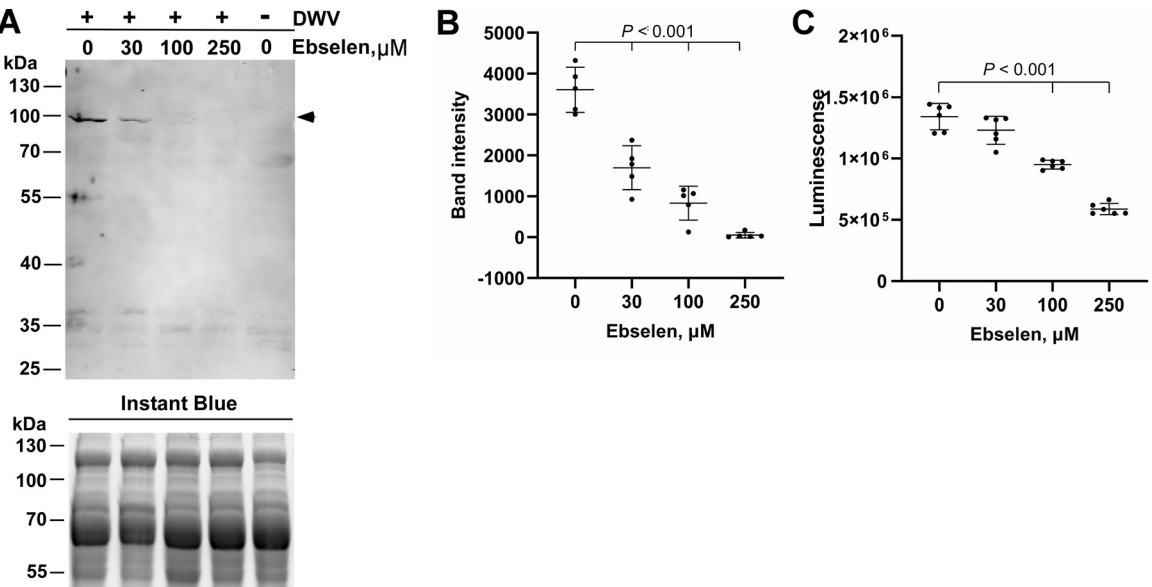

**FIG 3** Ebselen suppresses RNA-dependent RNA polymerase (RdRP) synthesis in the deformed wing virus (DWV)-infected pupal head cells. (A) Effect of increasing concentrations of ebselen on RdRP synthesis in DWV-infected (+) pupal head cells. Dimethyl sulfoxide (DMSO) was used as the control. RdRP precursor with 3C protease (97 kDa) is indicated by a black arrowhead. This band was absent in DWV-uninfected (−) cells. The replicate gel was stained with Instant Blue to show that equal amount of protein was applied in each lane. Molecular weight (kDa) of the protein marker is indicated at the left. (B) Band intensity of the RdRP precursor was compared between DMSO and ebselen at the indicated concentrations by the Dunnett method (one-tailed). Mean values ± SD (error bars) are shown ($n = 5$). (C) Luminescence generated by the luciferase activity dependent on the intracellular ATP level was compared between the pupal head cells treated with DMSO or ebselen at the indicated concentrations by the Dunnett method (one-tailed). There was no significant difference between DMSO and 30 μM ebselen. Mean values ± SD (error bars) are shown ($n = 5$).

substrate-binding domains, including conserved Y2299 and H2324 domains (Fig. 4C and F). N2134 of DWV is also well conserved between the 3C$^{pro}$s of Iflaviruses, but it localizes far from the catalytic center (Fig. 4D). Nevertheless, we found that the protease activity of the N2134A mutant was reduced to approximately one-third of that of the wild-type protein (Fig. 4F). A previous study suggested that H2190 and D2225 constitute the catalytic triad together with C2307 (14). However, H2190 was far from C2307 (Fig. 4E), and the H2190A mutant and wild-type protein showed comparable protease activity (Fig. 4F). D2225 is close to N2227 as described above, but the protease activity of D2225A was reduced to only half of that of the wild-type protein (Fig. 4F). Thus, H2190 and D2225 are not part of the catalytic triad of DWV 3C$^{pro}$. D2304A and E2329A mutants have the same activity as the wild-type protein, indicating that they are not the third residue of the catalytic triad (Fig. 4E and F). The above results are consistent with the structure of DWV 3C$^{pro}$ predicted by AlphaFold2.

We next determined how rupintrivir binds DWV 3C$^{pro}$ to inhibit its activity using a molecular docking tool. Rupintrivir was predicted to interact with 16 amino acids in the catalytic triad (C2307, H2170, and N2227) as well as a substrate-binding domain. Nitrogen atoms in the side chains of N2227 and H2324 appeared to form hydrogen bonds with rupintrivir. Five amino acids (I2283, N2284, A2285, L2288, and Y2289) in β-ribbon also appeared to interact with rupintrivir (Fig. 5A, B, and E). Thus, 3C$^{pro}$s of DWV and picornaviruses appear to bind rupintrivir in a similar manner (15, 37, 38). Analysis of rupintrivir binding with DWV 3C$^{pro}$ revealed a precise map of the amino acids critical for protease activity at the protein surface. The catalytic triad (C2307, H2170, and N2227) are adjacent to each other; R2156 and Y2171 are adjacent to C2307 and H2170. The largely reduced protease activities of R2156A and Y2171A mutants were consistent with their localization (Fig. 4F). The substrate-binding domain is composed of two grooves on the left and right sides, and V2325 and A2326 are in the middle to connect them. The left groove was constructed using H2302, G2303, D2304, G2305, H2324, G2327, and E2329. Among them, H2302,

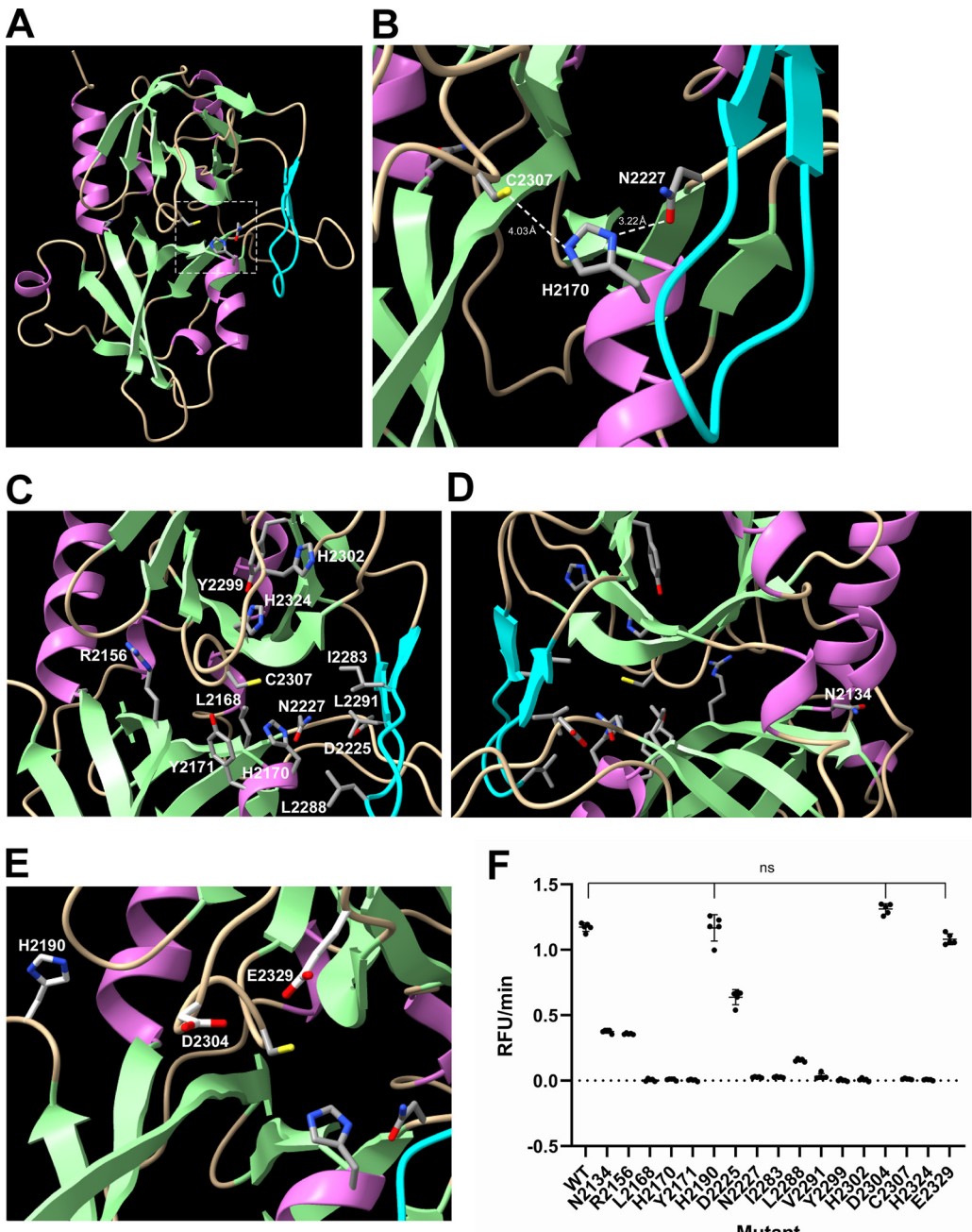

**FIG 4** Structure–function analysis of the deformed wing virus (DWV) 3C protease (3C^pro). (A) Structure of DWV 3C^pro predicted by AlphaFold2. The helix, strand, and coil structures are indicated by orchid, pale green, and wheat colors, respectively. β-ribbon is colored by cyan. The catalytic Cys-His-Asn triad is indicated by a square with a white dotted line. (B) Close-up view of the catalytic C2307-H2170-N2227 triad with distance information. (C) Positions of amino acid residues critical for the protease activity. (D) The position of N2134 that was well conserved between 3C^pros of Iflaviruses is shown by rotating the image (C) by 180°. (E) The positions of H2190, D2304, and E2329, which are not essential for the protease activity, are shown together with the catalytic C2307-H2170-N2227 triad. (F) Protease activities of wild type (WT) 3C^pro and alanine substituted mutants. Compared to WT, all mutants showed decreased activities, except H2190A, D2304A, and E2329A (ns), as indicated by the Dunnett test (one-tailed, $n = 5$).

G2303, D2304, and E2329 create a porous structure. The deep right groove is made by R2169, E2173, I2283, N2284, A2285, L2288, Y2289, and V2291. Most of the amino acids were in the β-ribbon structure (Fig. 5C and D).

**Self-cleavage of GST-DWV 3C^pro.** When we purified the 56 kDa wild-type GST-DWV 3C^pro protein, a 30 kDa small band was co-purified. This band was absent in the

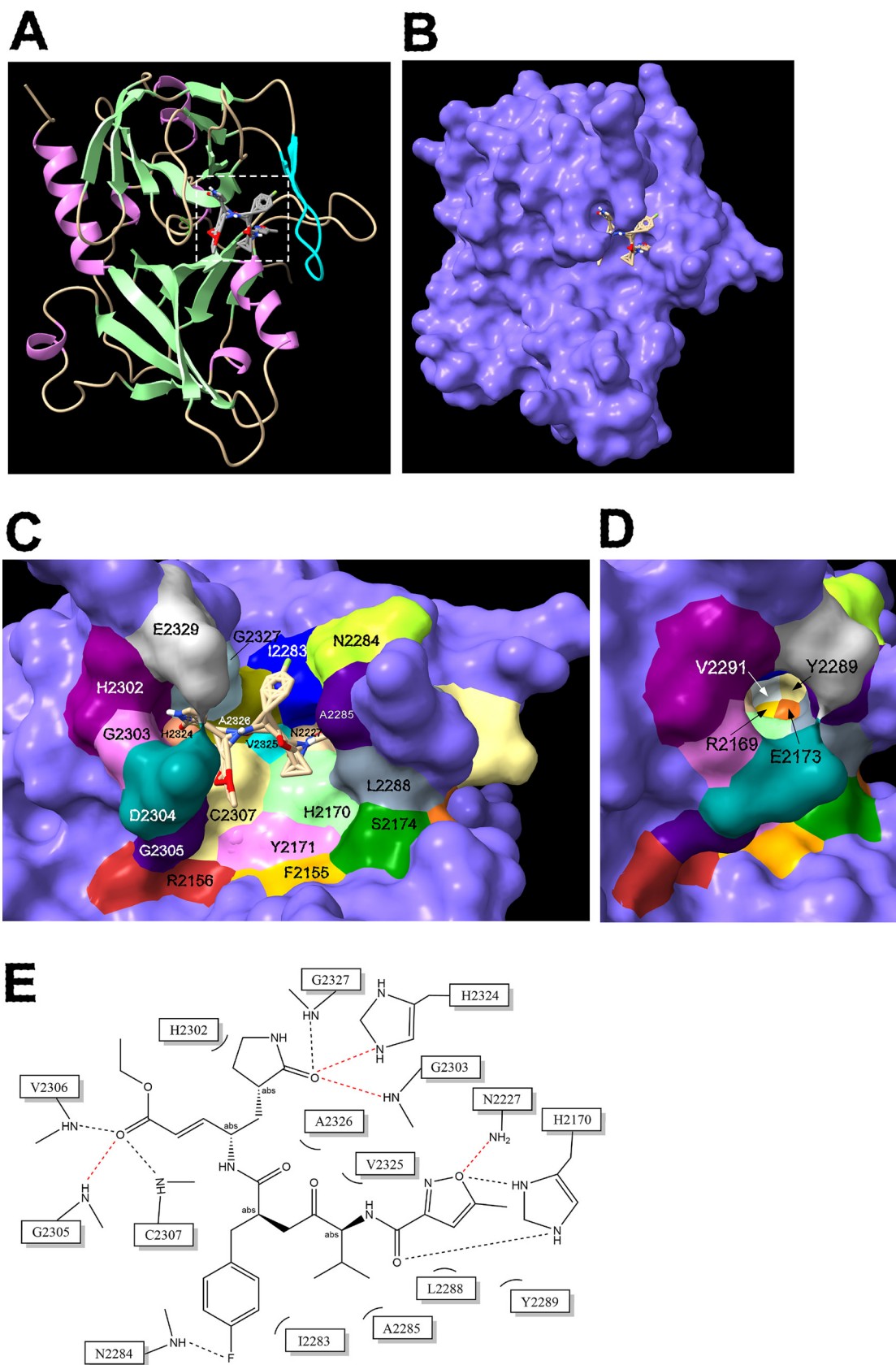

**FIG 5** Amino acids interacting with rupintrivir in deformed wing virus (DWV) 3C protease (3C$^{pro}$). (A) Overall structure of rupintrivir-bound DWV 3C$^{pro}$. Rupintrivir binding site is indicated by a square with white dotted line. (B) Surface view of

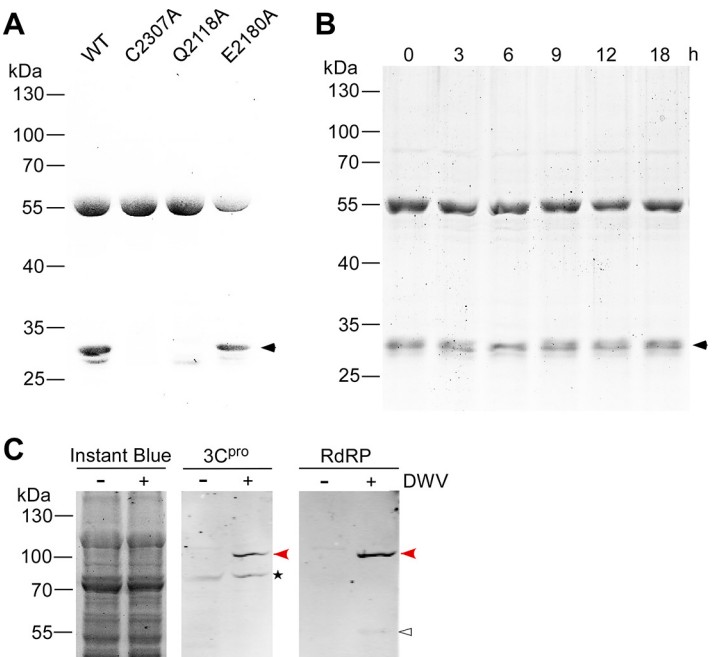

**FIG 6** Self-cleavage of the deformed wing virus (DWV) 3C protease (3Cᵖʳᵒ) and its synthesis in the virus-infected honey bee cells. (A) Sodium dodecyl sulfate-polyacrylamide gel electrophoresis (SDS-PAGE) of purified wild type (WT) and three mutant (C2307A, Q2118A, and E2180A) glutathione *S*-transferase (GST)-DWV 3Cᵖʳᵒ proteins. The 30 kDa band co-purified with 56 kDa GST-DWV 3Cᵖʳᵒ is indicated by a black arrowhead. Molecular weights (kDa) of the protein marker are at the left. (B) SDS-PAGE of purified GST-DWV 3Cᵖʳᵒ incubated at 33°C for the indicated time. (C) Western blotting of lysates of the control (−) or DWV-infected (+) honey bee pupal head cells by anti-3Cᵖʳᵒ (3Cᵖʳᵒ) or anti-RdRP (RdRP) antibody. The replicate gel was stained with Instant Blue to show that equal amount of protein was applied in each lane. Mature 42 kDa 3Cᵖʳᵒ and 55 kDa RdRP are indicated by black and white arrowheads, respectively. The 97 kDa precursor of 3Cᵖʳᵒ and RdRP (red arrowheads) was detected by both anti-3Cᵖʳᵒ and anti-RdRP antibodies. The above-mentioned three bands were specifically present in DWV-infected cells. Asterisk represents the nonspecific band detected by the anti-3Cᵖʳᵒ antibody.

mutant protein lacking protease activity, for example, C2307A (Fig. 6A). These results suggest that the 30 kDa band was generated by the self-cleavage of GST-DWV 3Cᵖʳᵒ either by *cis* or *trans* and contained GST. Our DWV 3Cᵖʳᵒ fusion protein contains two potential cleavage sites, Q2118 and E2180 (14). To determine the cleavage site, we prepared mutant proteins in which one of the above two amino acids was substituted with alanine. As shown in Fig. 6A, the 30 kDa band was absent only in the Q2118A mutant, suggesting that GST-DWV 3Cᵖʳᵒ is self-cleaved at Q2118 in *E. coli*. We also tested whether the purified GST-DWV 3Cᵖʳᵒ was self-cleaved during incubation at 33°C and found that it remained stable without self-cleavage for 18 h (Fig. 6B).

**Synthesis of 3Cᵖʳᵒ in DWV-infected honey bee cells.** To characterize the 3Cᵖʳᵒ synthesized in DWV-infected honey bee cells, we raised antibodies against DWV 3Cᵖʳᵒ. We

**FIG 5** Legend (Continued)
rupintrivir bound DWV 3Cᵖʳᵒ. The binding site of rupintrivir was predicted by the molecular docking tool, AutoDock Vina. (C) Amino acids interacting with rupintrivir as well as the neighboring ones are indicated by different colors. Rupintrivir is also shown. (D) Amino acids in the deep right groove of the substrate-binding domain created by β-ribbon are indicated by rotating the image (C) 45° without rupintrivir. (E) Sixteen amino acids interacting with rupintrivir. Hydrogen bonds are indicated by red dotted lines.

found that a 97 kDa protein specifically present in DWV-infected cells is recognized by both anti-3C[pro] and anti-RdRP antibodies, confirming that this is a 3C[pro] precursor with RdRP. There was also a 42 kDa band specifically recognized by the anti-3C[pro] antibody, suggesting that this corresponds to the mature 3C[pro] in DWV-infected cells (Fig. 6C). Thus, DWV 3C[pro] is present as both a precursor and a mature protein in the infected cells.

## DISCUSSION

Compared to the Km values of the best peptide substrate for 3C[pro]s of human rhinovirus (250 $\mu$M), poliovirus (7 $\mu$M), and enterovirus (30 and 43 $\mu$M) (21, 39–41), DWV 3C[pro] has a higher affinity for the peptide substrate used in this study (Km: 4.233 $\mu$M). Nevertheless, DWV polyprotein has other potential cleavage sites by 3C[pro] so that the efficiency of cleaving a peptide substrate containing these sites would be varied. The Kcat/Km value of DWV 3C[pro] was 59.96 mM$^{-1}$ min$^{-1}$ and comparable to those of hepatitis A virus (126 mM$^{-1}$ min$^{-1}$), enterovirus (11.8 and 0.71 mM$^{-1}$ min$^{-1}$), and foot-and-mouth disease virus (59.4 mM$^{-1}$ min$^{-1}$) 3C[pro]s with their best substrates (21, 41–43). These results suggest that the presence of Asn instead of Asp/Glu in the catalytic triad of DWV 3C[pro] does not dramatically affect protease activity. In fact, substituting Asp in the catalytic triad of hepatitis A virus 3C[pro] with Asn only resulted in slower processing (44).

Rupintrivir was the most effective to inhibit DWV 3C[pro] activity and the IC$_{50}$ value (0.36 $\mu$M) was lower than that of enterovirus 3C[pro] (1.65–7.3 $\mu$M) but higher than that of human rhinovirus 3C[pro] (5 nM) (37, 45, 46). Ebselen, disulfiram, and carmofur have been shown to inhibit the main protease of SARS-CoV-2 by covalently modifying cysteine in the catalytic diad (28, 29, 31). Although we were not able to predict the binding site of ebselen in DWV 3C[pro] using a molecular docking approach, it is likely to target C2307 in the catalytic triad to inhibit protease activity. Ebselen inhibited DWV replication in honey bee cells; however, it was also toxic to the cells (Fig. 3). Thus, ebselen (MW: 274.18) could be a potential lead compound for fragment-based drug discovery approaches to increase the specificity and potency to DWV 3C[pro]. Rupintrivir and quercetin (34) as well as ebselen inhibited DWV replication in cultured honey bee cells. Thus, these compounds and the other 3C[pro] inhibitors we identified in this study could be used to control DWV in honey bee colony.

In contrast to picornaviruses, DWV 3C[pro] contains Asn (N2227) as the third residue of the catalytic triad instead of Asp/Glu (Fig. 4A and B). Moreover, Asn is not conserved among Iflaviruses, and in fact other Iflaviruses and Dicistrovirus appear to have Asn/Ser/Glu/Asp residue (Fig. S2 and S3 in the supplemental material). The third residue of the triad could be flexible as long as it contained an oxygen in the side chain. Oxygen is necessary to maintain the entire architecture of the active site and stabilize the positively charged imidazole side chain of His during proteolysis (37). Meanwhile, Cys and His in the catalytic triad are under strong purifying selection, suggesting that they are the most critical amino acids for protease reaction. This is consistent with the fact that the main protease of coronavirus has the Cys-His catalytic diad (47). Because the N2227A mutant completely lost its protease activity (Fig. 4F), and the same was observed with other 3C[pro]s (48–50), the catalytic triad seems to be necessary for *Picornavirales*. In many Iflaviruses, the Tyr residue next to the His residue of the catalytic triad (Y2171 for DWV) is conserved and is substituted with Phe in the Dicistrovirus, CrPV. Thus, Tyr appears to be important to position the imidazole side chain of His close to the side chain of Cys by stacking interactions.

The pores made by H2302, G2303, D2304, and E2329 in the left groove of the substrate-binding domain did not appear to be essential because the protease activities of the D2304A and E2329A mutants were not affected (Fig. 4F). $\beta$-ribbon is the major component of the deep right groove of the substrate-binding domain. In picornavirus 3C[pro]s, the $\beta$-ribbon adopts an open conformation to increase the substrate accessibility to its binding domain, and the interaction between the $\beta$-ribbon and the N-terminal end of the substrate stabilizes the closed conformation to form an ES complex (15–21). Comparing the surface views of the substrate-binding domains of DWV, BrBV, LsPVI2, SeIV-1, SBV, and

CrPV 3C$^{pro}$ (Fig. 5C and D; Fig. S4 in the supplemental material), the overall conformation of the catalytic triad and $\beta$-ribbon was similar, but the shapes of substrate-binding domains were quite different. Thus, the structure of the substrate-binding domain of each 3C$^{pro}$ is shaped by a preferred cleavage site with a different amino acid sequence.

We found that GST-DWV 3C$^{pro}$ was self-cleaved at Q2118. However, this seems to only occur during the synthesis in *E. coli* because the mature folded protein was not self-cleaved at 33°C for 18 h (Fig. 6B). These results suggest that only the fraction of unfolded GST-DWV 3C$^{pro}$ is cleaved in *trans* at 15°C. It is likely that DWV polyprotein is cleaved at Q2118 to release the 97 kDa precursor of 3C$^{pro}$ and RdRP. However, we need to assume that the 97 kDa precursor is active as a protease similar to picornavirus 3CD precursor (51) and that cleavage should occur in *cis*, at least during the early stage of polyprotein synthesis until a sufficient 97 kDa precursor and 42 kDa mature 3C$^{pro}$s accumulate in DWV-infected cells. Because the 97 kDa precursor remains abundant in the virus-infected honey bee cells (Fig. 6C), processing between 3C$^{pro}$ and RdRP appears to be slower than the cleavage at Q2118. These characteristics appear to be shared among picornaviruses as well (52). Considering the size of mature 3C$^{pro}$ (42 kDa), the cleavage site should be beyond E2353 which was the C-terminal end of GST-DWV 3C$^{pro}$. It will be interesting to test whether the 97 kDa precursor and 42 kDa mature 3C$^{pro}$s have different protease activities and functions for viral RNA replication in future studies.

## MATERIALS AND METHODS

**Expression and purification of GST-DWV 3C$^{pro}$.** The 3C protease cDNA corresponding to amino acid 2094–2353 of DWV polyprotein was amplified by PCR using two primers, GST-3C$^{pro}$-5-BamHI and GST-3C$^{pro}$-3-NotI (Table S1). The amplified PCR product was digested by BamHI (NEB) and NotI (NEB), and then subcloned to pGEX-4T-1vector (Cytiva) followed by transformation to BL21. The transformed BL21 was grown in 1 L of LB medium containing 1% glucose and 0.1 mg/mL Ampillicin at 37°C until A$_{600}$ reached to 0.5. The cell suspension was cooled down, and then IPTG was added at 0.1 mM to induce the protein expression at 15°C for 16 h. *E. coli* was collected by centrifugation and resuspended in 100 mL of ice-cold TNED buffer (50 mM Tris-HCl, pH 8.0, 150 mM NaCl, 2 mM EDTA, 1 mM DTT) with 0.5% TX-100 and protease inhibitor cocktail (Beyotime). The cell lysate was prepared by sonication using Q700 Sonicator (Qsonica) at amplitude 100 on ice for 45 min (30 sec pulse with 3 min-interval). 1 mL of BeyoGold GST-tag Purification Resin (Beyotime) was added to the supernatant collected after centrifugation. After gently rotating at 4°C for 2 h, the resin was washed five times with 10 mL TNED buffer. The bound protein was eluted with 1. 5 mL TNED buffer containing 10 mM reduced glutathione. The two-thirds of eluted protein was dialyzed against 2L of TNED buffer twice at 4°C for 24 h. The rest of one-third was dialyzed against 2L of TNE buffer without DTT. We measured the protein concentration of purified protein using Enhanced BCA Protein assay kit (Beyotime).

GST-3C$^{pro}$ mutant proteins were generated by fusion PCR of two PCR products amplified using pGEX 5′ primer as well as the reverse primer for each mutant and pGEX 3′ primer as well as the forward primer for each mutant (Table S1). The plasmid DNA to express GST-DWV 3C$^{pro}$ was used as the template for the 1st PCR. The fusion PCR products were cloned in pGEX-4T-1vector and the mutant proteins were expressed and purified as above. All plasmid DNAs were sequenced to verify the intended mutations.

**Protease assay of GST-DWV 3C$^{pro}$.** Dabcyl-PVQAKPEMDNPNPGE-Edans derived from the cleavage site between L-protein and VP2 (underlined, 205-218 of polyprotein) was used as a peptide substrate to measure the protease activity. E was added to the C-terminus to link Edans. FRET experiments were performed with Varioskan LUX multimode microplate reader (Thermo Fisher). We measured 100 $\mu$L of reaction mixture with 50 mM Citrate buffer, pH 6.0, 150 mM NaCl, 2 mM EDTA, 2 mM DTT containing 1 $\mu$M GST-DWV 3C$^{pro}$ and 20 $\mu$M fluorogenic peptide substrate in a 96-well flat-bottom white microplate at 33°C. The relative fluorescence was measured using an excitation wavelength of 336 nm and by monitoring the emission at 490 nm every 3 min for 30 min. In order to determine the optimal reaction condition for GST-DWV 3C$^{pro}$, various pH values, temperatures, and DTT concentrations were used. To test the effect of DTT on the activity, the protein dialyzed against TNE buffer was used. To determine the velocity of product formation, we precalibrated the instrument with the free Edans standard in the presence of 20 $\mu$M Dabcyl to calculate the relationship between relative fluorescence and substrate concentration. We calculated the kinetic parameters, Km, Vmax, and catalytic efficiency (Kcat/Km) assuming Michaelis-Menten kinetics for the cleavage of peptide. No significant hydrolysis of the peptide substrates was observed in the absence of GST-DWV 3C$^{pro}$.

To identify the inhibitors for DWV 3C$^{pro}$, various concentrations of the tested compounds were added. To determine the inhibitory effects of the various compounds, we preincubated the compound and enzyme on ice for 10 min prior to addition of the peptide substrate. To analyze the effects of ebselen, disulfiram, and carmofur, we used the enzyme dialyzed against TNE buffer and carried out the reaction with or without DTT. We determined the initial velocities of the enzymatic reactions and fitted to a sigmoidal dose-response equation with nonlinear regression analysis using GraphPad Prism 9. The data from three independent assays were used to determine IC50 value of each compound.

**Infection of honey bee pupal head cells with DWV.** Honey bee pupae with pale/pink eyes were collected from a mite-free colony. They were surface sterilized by washing with 10% bleach followed by

sterile PBS three times (5 min for each wash). Heads from the pupae were dissected and homogenized seven times with 1 mL Grace culture medium using Dounce homogenizer (Loose fitting). The homogenate was then filtered through a cell strainer (Falcon) and the number of cells was counted. $10^6$ cells were suspended in 100 $\mu$L of Grace medium containing 10% FBS, antibiotics (penicillin and streptomycin), and ebselen at the indicated concentration with or without DWV (MOI: 10) in 24-well plate at 33°C for 1 h. DWV used for this study was one of type A strains (VD-B7) we previously isolated from *Varroa* mite-infested pupa (8). Fresh culture medium (400 $\mu$L) with ebselen at the indicated concentration was then added (the final cell density at $2 \times 10^6$/mL) and incubated at 33°C for 16 h.

**Western blot.** After DWV infection, the pupal head cells were collected by centrifugation, and then homogenized with 150 $\mu$L of RIPA buffer (20 mM Tris-HCl, pH 7.5, 150 mM NaCl, 1% NP-40, 0.5% sodium deoxycholate, 0.1% SDS) containing protease inhibitor cocktail. After centrifugation of the homogenate, the protein concentration of supernatant was measured using Enhanced BCA protein assay kit. The cell lysate with 20 $\mu$g of protein was analyzed by Western blot for each sample. The protein samples in SDS-PAGE sample buffer (2% SDS, 10% glycerol, 10% $\beta$-mercaptoethanol, 0.25% bromophenol blue, 50 mM Tris-HCl, pH 6.8) were heated at 95°C for 5 min. After centrifugation, the supernatants were applied to 10% SDS-PAGE and the proteins were transferred to a nitrocellulose membrane (Pall Life Sciences). Another 10% SDS-PAGE gel with the same samples was stained by Instant Blue. The membrane was then blocked with PBST (PBS with 0.1% Tween 20) containing 5% BSA at room temperature for 30 min followed by incubating with 1000-fold diluted anti-RdRP antibody (6) or 500-fold diluted anti-3C$^{pro}$ antibody at 4°C overnight. The membrane was washed five times with PBST (5 min each), and then incubated with 10,000-fold diluted IRDye 680RD donkey anti-rabbit IgG (H+L) (LI-COR Biosciences) in PBST containing 5% skim milk at room temperature for 2 h. The membrane was washed as above, and then visualized using Odyssey Imaging System (LI-COR Biosciences). Band intensity of 97 kDa RdRP precursor was measured by image-J.

**Testing cell viability.** Honey bee pupal head cells were prepared as described above and 100 $\mu$L cell suspension ($2 \times 10^6$/mL) was inoculated to each well in 96-well plate. The cells were cultured in the presence of either DMSO or ebselen at the indicated concentrations at 33°C for 16 h. The cultured plate was first incubated at room temperature for 10 min followed by adding 100 $\mu$L of reagent solution (CellTiter-Lumi Luminescent Cell Viability assay kit, Beyotime). The plate was shaken at room temperature for 2 min to promote cell lysis, and then further incubated for 10 min to stabilize the chemiluminescence signal. The signal was detected using Varioskan LUX multimode microplate reader and depends on intracellular ATP level, thus the relative viability of cells.

**Structural analysis of 3C proteases of DWV and other insect RNA viruses.** We determined the structures of 3C proteases of DWV, BrBV, LsPVI2, SelV-1, SBV, and CrPV using ColabFold (36) based on AlphaFold v2.1.0 (35). The generated structures were analyzed and displayed using UCSF ChimeraX (53, 54). The molecular structure of rupintrivir was collected from PubChem (CID: 6440352). We used AutoDock Tools package (55) to prepare the ligand and receptor by removing water and other heteroatoms and assigning partial charges. The size of the grid box for docking was set at 20 Å in each direction. We provided explicit coordinate to find all possible binding sites. After preparing the ligand, receptor, definition of binding site, AutoDock Vina (version 1.1.2) (56) program was used for the molecular docking simulation.

**Self-cleavage of GST-DWV 3C$^{pro}$.** Ten micrograms of wild type, C2307A, Q2118A, and E2180A GST-DWV 3C$^{pro}$ protein was analyzed by 10% SDS-PAGE followed by Instant Blue staining. 20 $\mu$g of wild-type GST-DWV 3C$^{pro}$ was incubated in the enzyme reaction buffer without the substrate at 33°C and the 4 $\mu$g protein was sampled every 3 h for 12 h and at 18 h. All samples including the one without the incubation (0 h) were analyzed by SDS-PAGE as above.

**Preparation of anti-DWV 3C$^{pro}$ antibody.** DWV 3C$^{pro}$ insert in pGEX-4T-1 was transferred to pGEX-6P-3 followed by transformation to BL21. The protein was expressed and purified as above except 5 mL TNED buffer containing 100 units of PreScission Protease (Beyotime) was added to the resin after the final wash. The resin was incubated at 4°C overnight and the supernatant with the released DWV 3C$^{pro}$ was collected. The remaining resin was eluted once with 5 mL TNED buffer, and the eluate was combined with the above supernatant. The total eluate was dialyzed against 2L of PBS with 1 mM EDTA, 0.5 mM DTT, and 0.1% sarcosyl twice at 4°C for 24 h. The purified protein was delivered to GeneScript-Nanjing to raise the anti-rabbit polyclonal antibody.

**Alignment of Ifravirus 3C protease sequences.** 3C protease sequences of various Iflaviruses were searched in NCBI by BLASTP using DWV 3C$^{pro}$ sequence as a query. Twenty-three sequences were picked up and aligned together with DWV 3C$^{pro}$ using MUSCLE program (EMBL-EBI).

**Statistical analysis.** All data presented were from representative independent experiments. Statistical analyses were performed with Bell Curve for Excel (Social Survey Research Information Co., Ltd.) and no data point was excluded. The applied statistical tests and *P*-values are described in figure legends.

## SUPPLEMENTAL MATERIAL

Supplemental material is available online only.

**SUPPLEMENTAL FILE 1**, PDF file, 0.6 MB.

## ACKNOWLEDGMENTS

We thank Yixiang Tang, Shiqi Xu, Jiarui Li, Zhuoran Leng, and Shangqing Li for their contributions to conduct the experiments. We are grateful to David Ruiz-Carrillo for his comments on the paper.

This work was supported by Jinji Lake Double Hundred Talents Program and XJTLU Summer Undergraduate Research Fellowship (SURF 2021095) to T.K.

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
