## [Reviewer comments · Microbiology Spectrum]

Microbiology Spectrum

DWV 3C protease uncovers the diverse catalytic triad in insect RNA viruses

Xuye Yuan and Tatsuhiko Kadowaki

Corresponding Author(s): Tatsuhiko Kadowaki, Xian Jiaotong-Liverpool University

Review Timeline:

Submission Date:	January 7, 2022
Editorial Decision:	February 28, 2022
Revision Received:	March 5, 2022
Editorial Decision:	April 13, 2022
Revision Received:	April 15, 2022
Accepted:	April 22, 2022

Editor: Frederick S. Kibenge

Reviewer(s): Disclosure of reviewer identity is with reference to reviewer comments included in decision letter(s). The following individuals involved in review of your submission have agreed to reveal their identity: Eugene V Ryabov (Reviewer #1)

Transaction Report:

DOI: <https://doi.org/10.1128/spectrum.00068-22>

February 28, 2022

Prof. Tatsuhiko Kadowaki
Xian Jiaotong-Liverpool University
Suzhou
China

Re: Spectrum00068-22 (DWV 3C protease uncovers the diverse catalytic triad in insect RNA viruses)

Dear Prof. Tatsuhiko Kadowaki:

This manuscript reports use of AlphaFold2 to predict the 3-dimensional structure of the 3C proteases of several iflaviruses, including Deformed wing iflavirus (DWV). The 3C protease is important in the life cycle of iflaviruses as it is required for the cleavage of the viral polyprotein to synthesize mature viral protein of these viruses. Inhibitors of this enzyme can be used to control DWN, a major cause of disease in honey bees. The 3-D structure allowed identification of the potential catalytic site of the DWV 3C Protease and model binding of potential inhibitors of the protease activity. The recombinant DWV 3C protease expressed in *E. coli* and the peptide corresponding to DWV LP-VP2 interface were then used to devise an in vitro FRET protease assay that allowed biochemical characterization of the DWV 3C protease.

This manuscript has been reviewed by two reviewers. Both reviewers recommend modifications by addressing several points.

Link Not Available

Sincerely,

Frederick S. Kibenge

Journals Department
Reviewer comments:

Reviewer #1 (Comments for the Author):

In this study, 3 dimensional structure of the 3C proteases of several iflaviruses, including Deformed wing iflavivirus (DWV), was predicated using Alphafold2. This allowed to identify potential amino acids in the catalytic site of the DWV 3C Protease and model binding of potential inhibitors of the protease activity. The recombinant DWV 3C protease expressed in *E. coli* and the peptide corresponding to DWV LP-VP2 interface were used to devise in vitro FRET protease assay. This assay has allowed to biochemically characterize DWV 3C protease and, using mutant versions of DWV 3C protease with alanine residues replacing the catalytic positions, to experimentally confirmed the conserved catalytic triad and mechanisms of the protease activity inhibition with known inhibitors, such as rupintrivir. The MS is in general well written, the methods are described in sufficient detail, and the results are novel. The described in vitro system fo retesting DWV 3C protease activity has a potential to be used for screening novel antiviral compounds with a potential to control DWV. I recommend to accept after addressing some minor points.

- Page 4. Include a section of expression and self-processing of the 56 kDa of the GST-DWV 3C Protease fusion as a first section of the Results section.

- Questions to the DWV 3C protease size"

- Page 6."When we purified the 56 kDa wild-type GST-DWV 3Cpro protein, a 30 kDa small band was co-purified. This band was absent in the mutant protein lacking protease activity, for example, C2307A (Fig. 6A)."

- Can the the 30 kDa band (Fig, 6 A, B) be the DWC 3C protease?

Explain why in Fig. 6 the proposed 3C protease has a weight of 42 kD (Fig 6C), but the recombinant is only 30 kD (Fig. 6 AB). Does it mean that the section of DWV polyprotein (positions 2094-2353, which will give approximately 30 kDa peptide) is shortened at the C-terminus that compared to the wild type 3C protease?

- Fig. 6 C. Mark position of the putative 97 kDa DWV 3C protease - RdRpol precursor.

- Page 3.

"DWV belongs to Iflavirus (a sister species of Dicistrovirus) in the order Picornavirales." change to "DWV belongs to the genus Iflavirus in the order Picornavirales."

- Page 4 (... a 15 amino acid peptide with the potential cleavage site, AKPEMD...)

Provide entire sequence of the 15 Amin acid peptide (PVQAKPEMDNPNPG) which was actually used in the assay. It was not shown if a short 6 amino acid peptide AKPEMD will act a s good substrate for the DWV 3C Protease.

This proteolytic site is not "potential" - it was experimentally demonstrated that a an insertion in the DWV polyprotes (e.g. GFP) which was be flanked by PVQAKPEMDNPNPG was completely excised in the course of DWV infection (see Ryabov <https://doi.org/10.3390/v12040374>)

Reviewer #2 (Comments for the Author):

Dear authors,

After reviewing your manuscript entitled "DWV 3C protease uncovers the diverse catalytic triad in insect RNA viruses", I have decided to suggest a re-submission after revisions.

I think the results of your work could be of interest to the readership of Microbiology Spectrum. However, the overall structure and flow of the manuscript should be enhanced to improve its clarity, and I provided below a series of general comments and suggestions I think will be needed to improve these points.

—

First, the absence of line numbers complicates significantly the reviewing process. Please add some in case you resubmit a next manuscript version.

—

Abstract

The objectives of the study are not clear: why did you focus on 3CPro and why is this specific enzyme important? Why did you compare it with picornaviruses?

The results of the current study should be better highlighted. At the moment, it is hard to say whether you only focused on DWV (as mentioned on the third sentence), or also analyzed other viruses (as mentioned later on)?

Introduction

Given that you submitted your work to a journal with broad readership interested in microbial ecology, I would suggest starting the introduction with the general information about viruses and the role of 3CPro, and then focus on honey bees. At the moment, the plan is not clear as the text shifts from hosts (paragraph 1), to viruses (paragraph 2) and then host again (paragraph 3).

First paragraph:

First sentence: no references are provided about *Tropilaelaps* sp. here.

Second sentence: the host = what host?

Second paragraph:

Fourth sentence: "well characterized in picornaviruses": as you focus on the host in the first paragraph, it might be worthwhile mentioning what host these viruses infect here too? Alternatively, moving this paragraph first might enhance clarity (see above).

Third paragraph:

It is not clear at this stage why there is a need to study what you did (e.g., structure-function, protease activities of mutant proteins). Why is this study important (e.g., how will it help controlling DWV in honeybee colonies)?

Results

As the methods are last, it might be helpful for the reader to add some explanations about why each experiment was conducted at the beginning of each subsection.

Enzymatic properties of DWV 3C

Why is it important to know the temperature and PH?

Inhibitors of DWV 3C

"Rupintrivir was the most effective compound and the IC50 value (0.36 μ M) was lower than that of enterovirus 3Cpro (1.65-7.3 μ M) but higher than that of human rhinovirus 3Cpro (5 nM) (Tan et al., 2016; Wang et al., 2011; Dragovich et al., 1999)." I would avoid providing citations in the results section, this relates more to the discussion.

Predicted structure of DWV 3Cpro by AlphaFold2

"We obtained multiple models that are very similar to each other": this is a bit vague, please provide details or remove.

Discussion

Before starting with the details, I suggest starting this section by summarizing the aim and major findings of the study in a few sentences.

In general, the discussion and results parts are very similar. Citations are provided in the results, and the discussion is providing many details. I miss the bigger picture in the discussion, why was this study conducted and what are the major findings. For instance, as the introduction is focused on honey bee health, I would expect more details about how the findings can help improving their health.

Methods

This section should provide more details about the origin of the samples and viruses analyzed. For instance, where do the viruses used come from for the "Expression and purification" part?

Supplementary

Do you plan to submit the relevant information on public repositories?

Staff Comments:

Preparing Revision Guidelines

Please return the manuscript within 60 days; if you cannot complete the modification within this time period, please contact me. If you do not wish to modify the manuscript and prefer to submit it to another journal, please notify me of your decision immediately so that the manuscript may be formally withdrawn from consideration by Microbiology Spectrum.

Reviewer #1 (Comments for the Author)

- Page 4. Include a section of expression and self-processing of the 56 kDa of the GST-DWV 3C Protease fusion as a first section of the Results section.

Authors: Since this section requires to compare between wild type and the protease dead mutant proteins, we think it is more logical to explain after describing the activities of various mutant proteins. Thus, it remains at the same position in the revised manuscript (line 193-204).

- Questions to the DWV 3C protease size"

- Page 6."When we purified the 56 kDa wild-type GST-DWV 3C_{pro} protein, a 30 kDa small band was co-purified. This band was absent in the mutant protein lacking protease activity, for example, C2307A (Fig. 6A)."

- Can the the 30 kDa band (Fig, 6 A, B) be the DWC 3C protease?

Explain why in Fig. 6 the proposed 3C protease has a weight of 42 kD (Fig 6C), but the recombinant is only 30 kD (Fig. 6 AB). Does it mean that the section of DWV polyprotein (positions 2094-2353, which will give approximately 30 kDa peptide) is shortened at the C-terminus that compared to the wild type 3C protease?

Authors: Because the expressed protein was affinity-purified using glutathione beads (BeyoGold™ GST-tag Purification Resin), the purified protein should contain GST (26 kDa). Therefore, the 30 kDa band can not be DWV3C^{pro} and is GST with 2094-2118 of DWV polyprotein. We expressed 2094-2353 of DWV polyprotein as 3C^{pro} (approx. 30 kDa) based on the homology to known 3C proteases of picornaviruses. However, as pointed out by the reviewer, the size of mature DWV3C^{pro} present in the infected cells is 42 kDa. This result suggests that cleavage between 3C^{pro} and RdRP occurs beyond the position 2353. These have been described in the revised manuscript (line 196-198 and 286-287).

- Fig. 6 C. Mark position of the putative 97 kDa DWV 3C protease - RdRpol precursor.

Authors: The 97 kDa band has been marked in the revised Figure 6 and the figure legend has been modified accordingly (line 706-707).

- Page 3.

"DWV belongs to Iflavirus (a sister species of Dicistrovirus) in the order Picornavirales." change to "DWV belongs to the genus Iflavirus in the order Picornavirales."

Authors: This has been changed in the revised manuscript (line 65).

- Page 4 (... a 15 amino acid peptide with the potential cleavage site, AKPEMD...)

Provide entire sequence of the 15 Amino acid peptide (PVQAKPEMDNPNG) which was actually used in the assay. It was not shown if a short 6 amino acid peptide AKPEMD will act as a good substrate for the DWV 3C Protease.

This proteolytic site is not "potential" - it was experimentally demonstrated that an insertion in the DWV polyprotein (e.g. GFP) which was flanked by PVQAKPEMDNPNG was completely excised in the course of DWV infection (see Ryabov <https://doi.org/10.3390/v12040374>)

Authors: These have been addressed in the revised manuscript (line 96-99).

Reviewer #2 (Comments for the Author):

Dear authors,

First, the absence of line numbers complicates significantly the reviewing process. Please add some in case you resubmit a next manuscript version.

Authors: We have added line numbers in the revised manuscript.

Abstract

The objectives of the study are not clear: why did you focus on 3CPro and why is this specific enzyme important? Why did you compare it with picornaviruses?

The results of the current study should be better highlighted. At the moment, it is hard to say whether you only focused on DWV (as mentioned on the third sentence), or also analyzed other viruses (as mentioned later on)?

Authors: As described in the Abstract, 3C^{pro} is necessary for the cleavage of the polyprotein to synthesize mature viral proteins. Thus, it is one of the non-structural viral proteins essential for the replication. (line 22-23 of the revised manuscript). Because the structures and functions of 3C proteases have been best characterized with picornaviruses, they serve as the targets for comparison with DWV 3C^{pro}. As seen with the Results section, we primarily focused on DWV 3C^{pro} and found that it has Cys-His-Asn catalytic triad rather than Cys-His-Asp/Glu which is common between 3C proteases of picornaviruses. That led us to predict the structures of other I flaviruses and Dicistrovirus using AlphaFold2 to determine whether there is the diversity of catalytic triads in insect RNA viruses.

Introduction

Given that you submitted your work to a journal with broad readership interested in microbial ecology, I would suggest starting the introduction with the general information about viruses and the role of 3C^{Pro}, and then focus on honey bees. At the moment, the plan is not clear as the text shifts from hosts (paragraph 1), to viruses (paragraph 2) and then host again (paragraph 3).

Authors: In the first paragraph, we introduce about the negative impact of DWV on honey bee colony and health. In the second paragraph, we describe that DWV belongs to the genus I flavivirus in the order *Picornavirales* which contains 3C protease in common. We then explain about the essential roles of 3C protease for replication of picornavirus. In the third paragraph, we briefly describe our study to characterize the structure and function of DWV 3C^{pro} to give insight into the mechanism of viral replication. We believe this is logical order for the Introduction section.

First paragraph:

First sentence: no references are provided about *Tropilaelaps* sp. here.

Second sentence: the host = what host?

Authors: These have been addressed in the revised manuscript (line 55-56).

Second paragraph:

Fourth sentence: "well characterized in picornaviruses": as you focus on the host in the first paragraph, it might be worthwhile mentioning what host these viruses infect here too? Alternatively, moving this paragraph first might enhance clarity (see above).

Authors: Picornaviruses infect a wide range of vertebrates. This has been added in the revised manuscript (line 69).

Third paragraph:

It is not clear at this stage why there is a need to study what you did (e.g., structure-function, protease activities of mutant proteins). Why is this study important (e.g., how will it help controlling DWV in honeybee colonies)?

Authors: Although DWV has been best studied among honey bee viruses, very little is known about the mechanisms of infection and replication in the host cells. Since 3C protease is crucial for viral replication and well characterized with picornaviruses regarding the structure-function relationship and inhibitors, our study on DWV 3C^{pro} should give insight into the mechanism of viral replication. This has been described in the revised manuscript (line 89-91).

Results

As the methods are last, it might be helpful for the reader to add some explanations about why each experiment was conducted at the beginning of each subsection.

Enzymatic properties of DWV 3C

Why is it important to know the temperature and PH?

Authors: These are important to determine the optimum condition to measure the protease activity. This has been mentioned in the revised manuscript (line 100-102).

Inhibitors of DWV 3C

"Rupintrivir was the most effective compound and the IC₅₀ value (0.36 μ M) was lower than that of enterovirus 3C_{pro} (1.65-7.3 μ M) but higher than that of human rhinovirus 3C_{pro} (5 nM) (Tan et al., 2016; Wang et al., 2011; Dragovich et al., 1999)." I would avoid providing citations in the results section, this relates more to the discussion.

Authors: This sentence has been moved to the Discussion section in the revised manuscript (line 228-231).

Predicted structure of DWV 3C_{pro} by Alphafold2

"We obtained multiple models that are very similar to each other": this is a bit vague, please provide details or remove.

Authors: This has been removed in the revised manuscript (line 128-129).

Discussion

Before starting with the details, I suggest starting this section by summarizing the aim and major findings of the study in a few sentences.

Authors: These are already explained in the Abstract and Introduction sections so that we think it is better to avoid repeating them in the Discussion section.

In general, the discussion and results parts are very similar. Citations are provided in the results, and the discussion is providing many details. I miss the bigger picture in the discussion, why was this study conducted and what are the major findings. For instance, as the introduction is focused on honey bee health, I would expect more details about how the findings can help improving their health.

Authors: We have added the following sentences, “Rupintrivir and quercetin (Wu *et al.*, 2021) as well as ebselen inhibited DWV replication in cultured honey bee cells. Thus, these compounds and the other 3C^{pro} inhibitors we identified in this study could be used to control DWV in honey bee colony.” in the revised manuscript (line 239-242).

Methods

This section should provide more details about the origin of the samples and viruses analyzed. For instance, where do the viruses used come from for the "Expression and purification" part?

Authors: We did not use DWV for the experiments described in “**Expression and purification of GST-DWV 3C^{pro}**” section. This explains how a plasmid DNA to express GST-DWV 3C^{pro} in *E. coli* was constructed and how the protein was purified. However, as pointed out by the reviewer, we used DWV for the experiments described in “**Infection of honey bee pupal head cells with DWV**” section. Thus, this section has been modified with the origin of DWV we used in the revised manuscript (line 356-357).

Supplementary

Do you plan to submit the relevant information on public repositories?

Authors: Since we did not sequence any novel DNA/RNA sequences, the Supplementary information will not be deposited to public data base.

April 13, 2022

Prof. Tatsuhiko Kadowaki
Xian Jiaotong-Liverpool University
Suzhou
China

Re: Spectrum00068-22R1 (DWV 3C protease uncovers the diverse catalytic triad in insect RNA viruses)

Dear Prof. Tatsuhiko Kadowaki:

Thank you for your responses and revisions. The reviewers have more questions that need to be addressed in the revised version.

Link Not Available

Sincerely,

Frederick S. Kibenge

Journals Department
Reviewer comments:

Reviewer #1 (Comments for the Author):

Most of the questions were addressed in the revised version, but few minor points need correction:

L.200. "...As shown in Figure 1A, the 30 kDa band was absent only in the Q2118A mutant" should it be Figure 6A? Fig. 1A does not show any SDS PAAG results. Also, in Fig. 6A, C2307A does not have 30 kDa band.

L.89. "...Our study on DWV 3Cpr gives insight into the mechanism of viral replication and may help to control DWV."

This study experimentally demonstrated proteolytic activity of the proposed 3C protease of DWV, model its 3D structure, and identified catalytic amino acid in the active site. Be more specific how this information "may help to control DWV" - e.e. by developing specific inhibitors of DWV 3C protease activity?

Reviewer #3 (Comments for the Author):

The author should indicate that "the specific inhibitors of DWV 3Cpro could be used to control DWV infection in honey bees" is a prospect, but the statement in the manuscript seems it is a confirmed result.

Importance:

I think the objectives "To understand the mechanism of DWV replication in the host cells," seems indistinct.

Introduction:

PAGE3 LINE83: I think it is better to use the reason why you focus on the catalytic triad and enzymatic properties of DWV 3CPro as the intro of the statement of your work.

PAGE3 LINE86: "the structure-function relationship", the "-" is in an inconsistent format.

Results-Inhibitors of DWV 3Cpro

Figure 3: The cell viability was reduced at concentrations of 100 and 250 μ M, but the protein was still in equal amount (Figure. 3A)?

Page5 line157 & Page6 line173: "the structure of DWV 3Cpro predicted by AlphaFold2 is correct"-this statement seems overgeneralizing.

Page7 line196 & line201: (Fig. 1A) into (Fig. 6A)

Staff Comments:

Preparing Revision Guidelines

Please return the manuscript within 60 days; if you cannot complete the modification within this time period, please contact me. If you do not wish to modify the manuscript and prefer to submit it to another journal, please notify me of your decision immediately so that the manuscript may be formally withdrawn from consideration by Microbiology Spectrum.

The author should indicate that “the specific inhibitors of DWV 3Cpro could be used to control DWV infection in honey bees” is a prospect, but the statement in the manuscript seems it is a confirmed result.

Importance:

I think the objectives “To understand the mechanism of DWV replication in the host cells,” seems indistinct.

Introduction:

PAGE3 LINE83: I think it is better to use the reason why you focus on the catalytic triad and enzymatic properties of DWV 3C^{Pro} as the intro of the statement of your work.

PAGE3 LINE86: “the structure–function relationship”, the “–” is in an inconsistent format.

Results-Inhibitors of DWV 3Cpro

Figure 3: The cell viability was reduced at concentrations of 100 and 250 μ M, but the protein was still in equal amount (Figure. 3A)?

Page5 line157 & Page6 line173: “the structure of DWV 3Cpro predicted by AlphaFold2 is correct”-this statement seems overgeneralizing.

Page7 line196 & line201: (Fig. 1A) into (Fig. 6A)

Reviewer #1 (Comments for the Author):

Most of the questions were addressed in the revised version, but few minor points need correction:

L.200. "...As shown in Figure 1A, the 30 kDa band was absent only in the Q2118A mutant" should it be Figure 6A? Fig. 1A does not show any SDS PAAG results. Also, in Fig. 6A, C2307A does not have 30 kDa band.

Authors: Yes, it should be Figure 6A but not Figure 1A. These have been corrected in the 2nd revision (Line 182 and 186).

L.89. "...Our study on DWV 3Cpr gives insight into the mechanism of viral replication and may help to control DWV."

This study experimentally demonstrated proteolytic activity of the proposed 3C protease of DWV, model its 3D structure, and identified catalytic amino acid in the active site. Be more specific how this information "may help to control DWV" - e.e. by developing specific inhibitors of DWV 3C protease activity?

Authors: The sentence has been changed to "Our study on DWV 3C^{pro} gives insight into the mechanism of viral replication and may help to control DWV infection in honey bees using the specific inhibitor." in the 2nd revision (Line 83-85).

Reviewer #3 (Comments for the Author):

The author should indicate that "the specific inhibitors of DWV 3C^{pro} could be used to control DWV infection in honey bees" is a prospect, but the statement in the manuscript seems it is a confirmed result.

Authors: The sentence has been changed to "Furthermore, it would be possible to use the specific inhibitors of DWV 3C^{pro} to control DWV infection in honey bees in future." in the 2nd revision (Line 33-34).

Importance:

I think the objectives "To understand the mechanism of DWV replication in the host cells," seems indistinct.

Authors: The sentence has been changed to "To give insight into the mechanism of DWV replication in the host cells, we studied the structure–function relationship of 3C protease (3C^{pro}), which is necessary to cleave a viral polyprotein at the specific sites to produce the mature proteins." in the 2nd revision (Line 39-42).

Introduction:

PAGE3 LINE83: I think it is better to use the reason why you focus on the catalytic triad and enzymatic properties of DWV 3C^{Pro} as the intro of the statement of your work.

Authors: We have added the following two sentences, “Because 3C^{pro} of insect RNA virus has never been studied, we expressed and purified DWV 3C^{pro} as a glutathione-S-transferase (GST) fusion protein to characterize the protease activity and the inhibitors in this study.” (Line 75-78) and “We focused on characterizing the catalytic triad since it represents the most critical domain of picornavirus 3C^{pro}.” (Line 81-82) in the 2nd revision.

PAGE3 LINE86: "the structure-function relationship", the "-" is in an inconsistent format.

Authors: These have been corrected in the 2nd revision (Lines 40, 78, and 642).

Results-Inhibitors of DWV 3C^{pro}

Figure 3: The cell viability was reduced at concentrations of 100 and 250 μ M, but the protein was still in equal amount (Figure. 3A)?

Authors: As described in the Materials and Methods section of manuscript (Line 341-342), we analyzed 20 μ g of total protein for each sample so that the band intensity stained by Instant Blue for each lane in Fig. 3A should be equal.

Page5 line157 & Page6 line173: "the structure of DWV 3C^{pro} predicted by AlphaFold2 is correct"-this statement seems overgeneralizing.

Authors: We have changed the statements to the followings, “To further examine the structure of DWV 3C^{pro} predicted by AlphaFold2, we prepared mutant proteins in which the potentially critical amino acid was substituted with alanine and measured the protease activities.” (Line 145-147) and “The above results are consistent with the structure of DWV 3C^{pro} predicted by AlphaFold2.” (Line 160-161) in the 2nd revision.

Page7 line196 & line201: (Fig. 1A) into (Fig. 6A)

Authors: These have been corrected in the 2nd revision (Line 182 and 186).

April 22, 2022

Prof. Tatsuhiko Kadowaki
Xian Jiaotong-Liverpool University
Suzhou
China

Re: Spectrum00068-22R2 (DWV 3C protease uncovers the diverse catalytic triad in insect RNA viruses)

Dear Prof. Tatsuhiko Kadowaki:

This manuscript is now acceptable for publication in Microbiology Spectrum.

Your manuscript has been accepted, and I am forwarding it to the ASM Journals Department for publication. You will be notified when your proofs are ready to be viewed.

Sincerely,

Frederick S. Kibenge
Editor, Microbiology Spectrum
